# Velocity Vector Estimation of Two-Dimensional Flow Field Based on STIV

**DOI:** 10.3390/s23020955

**Published:** 2023-01-13

**Authors:** Jianghuai Lu, Xiaohong Yang, Jianping Wang

**Affiliations:** Faculty of Information Engineering and Automation, Kunming University of Science and Technology, Kunming 650500, China

**Keywords:** hydrometry, velocity measurement, STIV, MOT, FMAA

## Abstract

As an important part of hydrometry, river discharge monitoring plays an irreplaceable role in the planning and management of water resources and is an essential element and necessary means of river management. Due to its benefits of simplicity, efficiency and safety, Space-Time Image Velocimetry (STIV) has attracted attention from all around the world. The most crucial component of the STIV is the detection of the Main Orientation of Texture (MOT), and the precision of detection directly affects the results of calculations. However, due to the complicated river flow characteristics and the harsh testing environment in the field, a large amount of noise and interfering textures show up in the space-time images, which affects the detection results of the MOT. In response to the shortage of noise and interference texture, a new non-contact image analysis method is developed. Firstly, Multi-scale Retinex (MSR) is proposed to pre-process the images for contrast enhancement; secondly, a fourth-order Gaussian derivative steerable filter is employed to enhance the structure of the texture; next, based on the probability density distribution function and the orientations of the enhanced images, the noise suppression function and the orientation-filtering function are designed to filter out the noise to highlight the texture. Finally, the Fourier Maximum Angle Analysis (FMAA) is used to filter out the noise further and obtain the clear orientations to achieve the measurement of velocity and discharge. The experimental results show that, compared with the widely used image velocimetry measurements, the accuracy of our method in the average velocity and flow discharge is significantly improved, and the real-time performance is excellent.

## 1. Introduction

China has many rivers and abundant water resources, hydrometry is an important basic work in our country, which is of great significance to the economic development of China. As an important part of hydrometry, river discharge monitoring plays an irreplaceable role in the planning and management of water resources, flood control and drought relief, and is an essential element and necessary means of river management [1]. In the 1980s, most of the hydrological monitoring stations used rotating-element current meters such as cup-type current meters or propeller current meters for the calculation of velocity and used the velocity-area method for the calculation of discharge. Overseas, Japan used the floats to measure the flow velocity and discharge during floods. However, the floats method became unreliable in case of large-scale floods and had the disadvantages of a lengthy observation time, a limited number of observations, and high risk for observers. The contact measurement is time-consuming and has strict environmental criteria for flow measurement. Therefore, it is essential to develop a non-contact method for quickly determining flow velocity.

For non-contact flow measurement techniques, the representative ones are Acoustic Doppler Current Profiler (ADCP) [2], particle image velocimetry (PIV) [3], large-scale particle image velocimetry (LSPIV) [4], feature matching velocimetry (FMV) [5], optical flow (OF) [6], and space-time image velocimetry (STIV) [7,8,9,10,11], the last five of which is river surface image velocimetry that uses the movement of artificially added tracer particles or visible "natural" features on the water surface (e.g., eddies, ripples, or other floats) to achieve non-contact measurement of surface velocity distributions. Surface velocity data can be gathered for extended periods of time using image-based velocimetry without the need for hydrographic workers to be on duty. The flow discharge is estimated in conjunction with the cross-sectional conditions once the surface velocity has been determined.

ADCP is a method of obtaining flow velocity by sending sound waves into the water and analyzing the scattered back signal for the Doppler effect. High standards for the flow measuring environment are required by ADCP, however there are no guarantees for the safety of the operators during extreme occurrences like floods. PIV is a method to measure the instantaneous flow field in two or three dimensions, mostly used for experimental analysis in indoor water tanks or wind tunnels, etc. It is one of the most effective tools to study the flow field and is mostly used for flow velocity analysis in small indoor areas (<50 cm${}^{2}$). Fujita et al. [12] improved the PIV technique for large-scale water surface flow field measurements under natural lighting conditions, using floating debris such as plant pieces or river surface flow features such as ripples and waves as tracers, and named it as large-scale particle image velocimetry (LSPIV). As an improvement of PIV, the biggest difference between LSPIV and traditional PIV is that LSPIV acquires video images under natural lighting conditions in the field from a small depression angle. LSPIV has the advantages of a larger measurement scale (riverine model: 1∼50 m; natural river: 100∼1000 m), simplified hardware equipment, and reduced tracer pollution to the river. LSPIV method has been employed since it was proposed, hydrologists have shown a lot of interest in it, and both domestic and international researchers have applied it to discharge measurements at hydrological stations. However, there are some shortcomings such as time-consuming calculation of cross-correlation coefficients, complicated parameter setting, large storage space required, low computational efficiency, and low spatial resolution of flow field. Zuniga [13] and Cao [14] proposed feature matching velocimetry (FMV), which is a feature points tracking algorithm in the field of computer vision introduced into the flow field measurement to extract and match the water surface feature points to characterize the motion of the water, and verified the applicability of the method. Cao et al. [5] applied FMV to weir flow and jet flow experiments, the output velocity fields has high temporal–spatial resolution, which truly reflected the river surface flow characteristics under the test conditions. Compared with the classical PIV and LSPIV algorithms, the FMV algorithm has the advantages of high accuracy, high computational efficiency, high temporal–spatial resolution, and less input parameters. However, the two computationally intensive and complex procedures of feature points extraction and description, matching and error detection in the FMV method will have a direct impact on the results of the measurement. The optical flow method uses the change of pixels in the time domain and the correlation between two frames to calculate the motion information of the objects between the two frames. Traditional optical flow methods cannot analyze complex flow situations well, so many researchers have improved optical flow to improve computational accuracy. Tauro et al. [15] used the differential sparse Lucas–Kanade algorithm to track FAST, ORB, SIFT, SURF and GFTT based on the optical tracking velocimetry (OTV), and filtered the tracking trajectories to retain the trajectories related to the actual objects in the field of view. It is proven that the method can quickly and reliably estimate the surface flow velocity by putting it to the test on two sets of picture data collected under various natural situations. However, the selection of feature points is arbitrary and lacks the theoretical practical proof, as well as high computational complexity and low efficiency for tracking hundreds of thousands of trajectories. Khalid et al. [6] present a weighted diffusion term to compensate for small scale contributions based on the scalar transport equation. Lu et al. [16] proposed the field-segmentation-based variational optical flow (FS-VOF) to preserve the spatial discontinuity property of the non-uniform flow field. Its data term is based on the partitioned region and is derived from the physically based optical flow equation. Both optical flow methods are less robust to illumination and has some limitations for large displacement estimation. The space-time image velocimetry (STIV) method is a non-contact velocimetry method that uses the velocity-measuring line as the analysis region, and estimates the one-dimensional time-averaged flow velocity by detecting the Main Orientation of texture (MOT) of the space-time image (STI). It has been successfully used in China, Japan, Australia, France, and South Africa to measure the velocity and discharge during floods. The STIV consists of three main components: space-time image generation, the MOT detection, velocity and discharge calculation. Among them, detection of the MOT is the most important part, the accuracy directly determines the calculation results of discharge, which is the core and difficult part of this method. To calculate the MOT, researchers have proposed the deformation method [17], luminance gradient tensor method (LGTM) [18], two-dimensional auto-correlation function (QESTA) [19], Fourier maximum angle analysis (FMAA) [20], and deep learning (DL) [21]. The deformation method is to find the angle that minimizes the cumulative value of the absolute difference in vertical brightness, which has the problem of poor detection accuracy and is less applied at present. The LGTM uses the tensor analysis theory for the detection of the MOT, and in order to reduce the influence of noise as much as possible, the STI is divided into several subregions, using the coherency C to assess whether the texture is clear or unclear, subregions with clear orientation has higher values and anisotropic gray-level structure has lower values, but when the interfering texture and noise components in the images are relatively large, it will produce large errors or even wrong results. The QESTA uses the standardization filter to equalize the image intensity of the uneven distribution in the image and normalizes the image by the standard deviation of the vertical pixel array, and the texture direction is clearer, but it can only eliminate the interfering texture in the vertical direction, which is also sensitive to noise. The FMAA first filters out part of the interfering texture and noise through pre-processing, and then further filters out the noise through filtering in the frequency domain to obtain images with high clarity and low noise, which significantly improves the detection accuracy of the Main Orientation of the Spectra (MOS). The pre-processing is crucial for the subsequent detection of the MOT, which directly affects the accuracy. The DL constructs synthetic STI datasets and natural river STI datasets and uses the powerful learning ability of neural networks for training to obtain accurate computed values of MOT. Li et al. [22] used the residual network to construct a regression model to automatically extract effective texture features and learn the mapping function from image-space to angle-space to obtain the computed value of the MOT. However, due to the large number of datasets, the lack of pre-processing as a key process leads to a lack of measurement robustness and accuracy needs to be improved. In addition, the datasets tend to select the STI generated in good scenes, lacking in more complex changing scenes, and the dataset needs to be further expanded.

In order to improve the texture orientation information of the STIs and make them clearer to improve the accuracy, firstly, a multi-scale Retinex (MSR) is designed to preprocess the images with contrast enhancement; secondly, based on the contrast enhancement, a fourth-order Gaussian derivative steerable filter is used to further process the images and effectively enhance the image texture orientation structure; next, based on the probability density distribution function and texture main direction, the noise suppression function and orientation-filtering function are designed to filter out the noise to highlight the texture direction information and reduce the adverse effects of noise; finally, the FMAA algorithm is used to further filter out the noise and obtain the texture orientation information, so as to realize the measurement of flow velocity and discharge.

## 2. Outline of the Space-Time Image Velocimetry

The frequent occurrence of flood disasters in recent years has made it obvious that it is important to accumulate basic hydrological data such as rainfall, water level and flow discharge, which are the basis to deal with river disasters. Non-contact automatic measurement systems are necessary and have a bright future, allowing continuous, multi-point, and accurate collection of water levels and flow discharge, even during floods. Image-based methods of river flow measurement are favored by hydrologists for their stability, safety, and economy. Among the image-based methods, the Space-Time Image Velocimetry (STIV) is considered a powerful tool for obtaining hydrologic information. The STIV is a time-averaged velocity measurement method, which takes river surface images as the analysis object and takes the single pixel-wide velocity-measuring line as the analysis region, and detects the MOT in a generated space-time image to obtain one-dimensional velocities of the water surface, which has higher spatial resolution and computational efficiency and is more suitable for shore-based systems with a small depression angle. Among five methods of STIV, the FMAA method converts the complex MOT detection in the spatial domain into detecting the MOS in the frequency domain, which simplifies the complex convolution or gradient operation in the spatial domain and provides higher robustness to random noise and interfering textures in space-time images, and the computational efficiency is improved accordingly.

## 3. Principle of the Space-Time Image Velocimetry

Without considering the effect of wind, when the water moves at a fast speed, the water surface forms small ripples, eddies or other flow features due to the influence of the fluctuations of the river surface, roughness and structures of the riverbed. Taking advantage of these features, we can visually observe the movement of the water, and at this time the flow velocity is approximately equal to the velocity of these features. During the movement of these features, the grayscale of the river surface also changes accordingly, and the magnitude of the change reflects the magnitude of the flow velocity.

### 3.1. Generation of Space-Time Images

Firstly, a sequence of *M* frames of river surface images is acquired at a time interval Δt; then, a series of velocity-measuring lines of single pixel width and *L* pixels length are set in the image along the direction of the movement of the water, and each velocity-measuring line corresponds to a time-averaged flow velocity; next, the grayscale of each velocity-measuring line is extracted frame by frame, and a space-time image of size L×M pixels is synthesized with x−t as Cartesian coordinate system. Figure 1 shows the schematic diagram of the generation of space-time image, where the horizontal direction of the image represents the distance (pixels) and the vertical direction represents the time (image frames), describing the relationship between time and space in the form of a two-dimensional image of the movement of flow features on the river surface. As shown in Figure 1, significant directional textures (trajectories with a certain contrast to the background) appear in the space-time image due to the change of grayscale, and the angle between the direction of the texture and the *t*-axis is defined as the main orientation of texture (MOT), the magnitude of MOT reflects the magnitude of the time-averaged flow velocity on the velocity-measuring line.

Figure 2 shows the schematic diagram of the principle of space-time image generation, showing the movement positions of the flow features in the river for three moments $\left(t_{i},t_{i}+T,t_{i}+2T\right)$, different flow velocities ($v_{1}$ and $v_{2}$). Due to $v_{1}>v_{2}$, the MOT $\theta_{1}>\theta_{2}$ in the generated space-time image, the velocity of the flow features and the angle of its texture in the space-time image are in mutual correspondence. Therefore, the tracking and velocity measurement of the flow features can be achieved by detecting the MOT in the space-time image. The key to flow velocity measurement is detecting the complex nonlinear data MOT with higher accuracy.

### 3.2. Detection of the MOT

Texture analysis occupies an important position in the field of computer vision and image processing. Especially in image-based measurement techniques, texture orientation is a very important feature in texture analysis, efficient and accurate acquisition of texture orientation information has become a new research hotspot in recent years.

The space-time images obtained based on the information extraction in image sequences have low contrast and a low signal-to-noise ratio between texture structure information and noise information, and the directional texture features are not obvious, which greatly affects the final measurement results. Therefore, before measuring the MOT, the space-time image needs to be pre-processed with texture enhancement, mainly including contrast enhancement and denoising. After pre-processing the space-time image, the space-time image is further processed by using the fan-type filter in the frequency domain to obtain a space-time image with low noise and clear texture direction information, to achieve the purpose of flow velocity measurement.

#### 3.2.1. Space-Time Image Filtering and Enhancement

Since the signal-to-noise ratio and contrast of the space-time images are low and the texture features are not obvious, the space-time images are first enhanced and filtered, including multi-scale Retinex (MSR), fourth-order Gaussian derivative steerable filter, noise suppression function and orientation-filtering function.

(i)Multi-Scale Retinex (MSR) enhancement

Multi-Scale Retinex (MSR) [23] is an algorithm developed based on the Single-Scale Retinex (SSR). The image is divided into illumination image L(x,y) and reflection image R(x,y), and the image perceived by the human eyes can be expressed by Equation (Equation 1): (1)I(x,y)=L(x,y)·R(x,y)
where, L(x,y) denotes the low frequency component contained in the image background; R(x,y) denotes the high frequency component of the image, which is expressed as the detail component of the image, that is the enhanced image to be obtained; I(x,y) denotes the original image perceived by the human eyes. If the illumination image is removed and the reflection image is decomposed, the image detail information can be obtained. Simplifying Equation (Equation 1) in the logarithmic domain, we can get Equation (Equation 2): (2)$$\log_{2}\left[R\left(x,y\right)\right]=\log_{2}\frac{I(x,y)}{L(x,y)}=\log_{2}I\left(x,y\right)-\log_{2}\left[I\left(x,y\right)*F_{n}\left(x,y\right)\right]$$
where, $F_{n}\left(x,y\right)$ denotes the center surrounding function. The illumination component is obtained by convolving the center surrounding function with the input image.
(3)$$F_{n}\left(x,y\right)=\frac{1}{2\pi \sigma }\exp\left\{\frac{-\left(x^{2}+y^{2}\right)}{2\sigma^{2}}\right\}$$

Thus, there are: (4)$$R\left(x,y\right)=\sum_{i=1}^{N}\omega_{k}\left\{\log_{2}I_{i}\left(x,y\right)-\log_{2}\left[I_{i}\left(x,y\right)*F_{n}\left(x,y\right)\right]\right\}$$
$\omega_{k}$ is the weight parameter, which indicates the value of the weight of the i layer, and $\sum_{i=1}^{N}\;w_{k}=1$.

(ii)Fourth-order Gaussian derivative steerable filter

The texture is further enhanced with a fourth-order Gaussian derivative steerable filter [24,25] on top of the MSR enhanced image.

The steerable filter bank with different rotation angles is denoted as $h_{\theta }\left(x,y\right)$, and the image R(x,y) is convolved with $h_{\theta }\left(x,y\right)$ to obtain the filtered and enhanced image as follows.
(5)$$F\left(x,y\right)=R\left(x,y\right)*h_{\theta^{*}\left(x,y\right)}$$

In the result of convolving $h_{\theta }\left(x,y\right)$ with R(x,y), the orientation corresponding to when the pixel has the maximum response is denoted as $\theta^{*}\left(x,y\right)$, then we have: (6)$$\theta^{*}\left(x,y\right)=\arg\max_{\theta }\left\{R\left(x,y\right)*h_{\theta }\left(x,y\right)\right\}$$

(iii)Noise filtering function

After the fourth-order Gaussian derivative steerable filtering enhancement process, the low-contrast texture is enhanced, but the noise is also enhanced simultaneously, and the space-time image can be regarded as an image composed of texture and noise together, so the noise needs to be suppressed and the texture is retained.

Assuming that $h_{\theta }\left(x,y\right)$ responses to noise obeys Gaussian distribution, the distribution function is denoted as p(v∣n), and the distribution function of the response to the texture is denoted as p(v∣t), the enhanced space-time image can be considered as consisting of the joint distribution function of the response to noise and the response to texture: (7)$$p\left(v\right)=\omega_{n}p\left(v|n\right)+\left(1-\omega_{n}\right)p\left(v|t\right)$$
where: (8)$$p\left(v|n\right)=\frac{1}{\sqrt{2\pi }\sigma_{n}}\exp\left\{-\frac{{\left(v-u_{n}\right)}^{2}}{2\sigma_{n}^{2}}\right\}$$$\omega_{n}$ is the weighting factor, $0<\omega_{n}<1$; $\sigma_{n}$ and $u_{n}$ are the mean and variance of the Gaussian distribution function, respectively. By analyzing the joint distribution function, the response belonging to the texture is given a higher weighting value, while the response belonging to the noise is given a smaller weighting value so that the texture can be retained while the noise is filtered out. The expression of the noise filtering function is given by
(9)$$g_{1}\left(v\right)=p\left(v|t\right)=1-\frac{\left(1-\omega_{n}\right)p\left(v|n\right)}{p(v)}$$
where, $0<g_{1}\left(v\right)\leq 1$. The filtered noise image is obtained by finding $\omega_{n}$, $\sigma_{n}$, $u_{n}$, through the log-likelihood function.

(iv)Orientation-filtering function

After the noise filtering process in part (iii), there is still a part of noise, and the direction of these noises is not the same as the main direction of the texture of the space-time image.

$H_{\theta }\left(x,y\right)$, the Hilbert transform of $h_{\theta }\left(x,y\right)$, there is: (10)$$H_{\theta }\left(x,y\right)=Hilbert\left\{h_{\theta }\left(x,y\right)\right\}$$

The directional energy of the image f(x,y) is denoted as $E_{\theta }\left(x,y\right)$, whose expression is given by: (11)$$E_{\theta }\left(x,y\right)=\sum \left\{{\left(f*h_{\theta }\right)}^{2}+{\left(f*H_{\theta }\right)}^{2}\right\}$$

The MOT $\theta_{\max}$ is: (12)$$\theta_{\max}=\arg\max_{\theta }\left\{E_{\theta }\left(x,y\right)|\theta \in \left(0,\pi \right]\right\}$$

Therefore, the orientation-filtering function is given by: (13)$$g_{2}\left(x,y\right)=\frac{1}{\sqrt{2\pi }\sigma }\exp\left\{-\frac{{\left\{\theta^{*}\left(x,y\right)-\theta_{\max}\right\}}^{2}}{2\sigma^{2}}\right\}$$
where, $0<g_{2}\left(x,y\right)<1$. The orientation information represented by the noise can be filtered out by using the orientation-filtering function to keep the directions close to $\theta_{\max}$ and remove the directions far from it.

#### 3.2.2. Enhancement and Filtering Processing Results

The results of space-time image preprocessing are shown in Figure 3. After the enhancement and filtering process, the main directions of the texture of the space-time image are clearer.

Figure 4 shows the enhancement and filtering preprocessing results of the space-time image by histogram equalization, standardization filtering, Hanning window function filtering and the new method. Due to the influence of environmental factors, the direction of the texture in the space-time image is unclear, the new preprocessing method enhances the orientation of the texture, which makes the subsequent detection of the main orientation of the texture much easier.

#### 3.2.3. Denoising Algorithm Based on Fan-Type Filter

After the image enhancement and filtering process, the texture features are obvious and the image orientation information is richer, but inevitably some noise still remains, which still has a negative impact on the extraction of texture orientation. For this reason, further processing is needed to filter out the remaining noise information.

For the obvious difference between image signal and noise signal, researchers have proposed two denoising methods, spatial-domain denoising and transform-domain denoising. Spatial denoising is a method of “averaging” or “smoothing” the image grayscale values to disperse the abrupt noise components to the surrounding pixels and reduce the impact of noise by smoothing the image. Transform-domain denoising converts the signal in the spatial-domain to the transform-domain, and then filters out the useless background noise signals by some method while preserving the image structure information as much as possible, and finally restores the image information by inversion, to obtain an image with low noise signal and rich structure information.

In the study of transform-domain denoising algorithms, the Fourier transform method is used more often. The Fourier transform converts the image from the gray space domain to the frequency domain of gray change. The frequency of the image in the frequency domain represents the degree of gray change in the image, and for a region with uniform gray distribution, the frequency is lower, while for some regions with edge, texture and other information, the corresponding frequency is higher. Therefore, the Fourier transform classifies and integrates the frequency information of these regions, which can be easily processed in the transform-domain and finally inverted to obtain the required results. Many scientific studies have proved that Fourier transform denoising methods can better remove the noise while preserving the structural information of image features.

The denoising algorithm based on fan-type filter is shown in Figure 5.

(i)Detection of the main orientation of the spectra

In the field of digital image processing techniques, the most used is the two-dimensional discrete Fourier transform (2D-DFT), as shown in Equation (Equation 14).
(14)$$F\left(u,v\right)=\sum_{x=0}^{M-1}\sum_{y=0}^{N-1}f\left(x,y\right)e^{-j2\pi \left(\frac{ux}{M}+\frac{y}{N}\right)}$$
where, (x,y) denotes the spatial variables, (u,v) denotes the frequency variables; f(x,y) denotes an image of size M×N, F(u,v) denotes a spectral image of the same size; *M* and *N* denote the height and width of the image, respectively; *x*, *y*, *u* and *v*, the size ranges are x=0,1,⋯,M−1, y=0,1,⋯,N−1, u=0,1,⋯,M−1 and v=0,1,⋯,N−1, respectively.

The two-dimensional Fourier transform is often expressed in the form of polar coordinates, as shown in Equation (Equation 15).
(15)$$F\left(u,v\right)=\left|F\left(u,v\right)\right|e^{j\Phi (u,v)}$$

Its corresponding spectrum is given by
(16)$$\left|F\left(u,v\right)\right|={\left[F^{2}\left(u,v\right)+R^{2}\left(u,v\right)\right]}^{\frac{1}{2}}$$

The 2D-DFT is performed on the digital image to be processed, and the spectrum image is obtained. The visualization effect is poor due to the uneven distribution of grayscale in the spectrum image and the large range of values. To compress the dynamic range and avoid losing the detail information, the spectrum image is log-transformed. After the logarithmic transformation, the spectrum image has more visible details, which is beneficial to the visualization of the spectrum image. After the above steps, the spectrum shown in Figure 5b is obtained. From Figure 5b, the energy distribution of the band texture in the spatial domain with certain directionality in the frequency domain is in the center of the image and in a straight line perpendicular to the main direction of the texture.

The main orientation of the spectra is orthogonal to the main orientation of the texture in the space-time image, so the measurement of the MOT can be reduced to the measurement of the MOS.

Firstly, establish the polar coordinate system ρOθ with *O* as the origin in the spectrum shown in Figure 5b. Then calculate the integral value along the radial direction for each angle in the semicircle space of 0${}^{\circ }$∼180${}^{\circ }$ with the step of $1^{\circ }$.
(17)$$\left|F\left(\theta \right)\right|=\int_{0}^{\max(\rho )}\left|F\left(\rho ,\theta \right)\right|d\rho$$

The angle θ that maximizes |F(θ)| is denoted as the MOS $\theta_{m}$: (18)$$\theta_{m}=\arg\max\left|F\left(\theta \right)\right|$$

(ii)Filter selection and related parameter settings

Most of the textures in the space-time image are superimposed along the same direction, which ideally should be a straight line passing through the center of the spectrum, however, because the texture is not a standard straight line and the slope of each line fluctuates within a certain range, so it is a texture with a certain width in the spectrum. To obtain the MOS, the angle of this texture needs to be measured precisely.

Since the fan-type filter [26,27,28] has the characteristics of high accuracy, low computational complexity in image texture orientation detection and can use the fan-type filter to retain flow-related area to filter out the non-meaningful distractor elements related to the interference texture and other background noise. Therefore, choose the fan-type filter to process the texture with a certain width, this results in a space-time image with high resolution and improves the detection accuracy.

As shown in Figure 6, the filters are the upper and lower sectors $F_{1}$ and $F_{2}$ symmetrical about the center of the origin of polar coordinates. A sensitivity analysis of the parameter settings of the fan-type filter in the frequency domain was done in [28], and it was found that the filter can effectively filter out the noise when the angle β is $10.6^{\circ }$. Since the Section 3.2.1 has reduced the noise to a certain extent, the angle β is set to $8^{\circ }$ to extract as much effective texture structure as possible, the angle β is set as shown in Equation (Equation 19): (19)$$\beta =\left\{\begin{array}{c} \left[\theta_{m}-4^{\circ },\theta_{m}+4^{\circ }\right],F_{1}\;\\ \left[\theta_{m}+176^{\circ },\theta_{m}+184^{\circ }\right],F_{2} \end{array}\right.$$

(iii)Create the masked image

After selecting the appropriate filter and setting the corresponding parameters, we further constrain the gray value of the fan-shaped area in the spectrum, set the value corresponding to the fan-shaped area to 1 and the other areas to 0, as shown in Equation (Equation 20): (20)$$M\left(u,v\right)=\left\{\begin{array}{c} 1,F_{1}\&F_{2}\\ 0,\;\mathrm{otherwise}\; \end{array}\right.$$

As shown in Figure 5c, the white region indicates the retained components, which represent the region where the texture signal is located, and its value is 1; the black area indicates the filtered components, which represent the region where the noise signal is located, and its value is 0.

(iv)Acquisition of low-noise space-time images

In order to acquire the space-time image with low noise in a display, the masked image M(u,v) is multiplied by bit with the image F(u,v), and then the Fourier inverse transform is used to recover the image and transform the image to the spatial domain, so as to obtain the image with low-noise content and clear texture direction, as shown in Figure 5d.
(21)$$F^{\prime }\left(u,v\right)=F\left(u,v\right)\times M\left(u,v\right)$$
(22)$$f^{\prime }\left(x,y\right)=\frac{1}{MN}\sum_{x=0}^{M-1}\sum_{y=0}^{N-1}F^{\prime }\left(u,v\right)e^{j2\pi \left(\frac{ux}{M}+\frac{vy}{N}\right)}$$
where, $f^{\prime }\left(x,y\right)$ denotes the denoised image with clear orientations.

### 3.3. Calibration

After the MOT is obtained, the motion vector in the flow field needs to be converted from phase plane coordinates to actual spatial Cartesian coordinates to find the actual distance represented by each pixel. Four ground control points (GCPs), which are in the same plane and coplanar with the horizontal plane, are selected on both sides of the river. The spatial Cartesian coordinates of the GCPs are obtained using an industrial-grade total station whose performance is stable. Thereby, the relationship between the phase plane coordinates and the actual spatial Cartesian coordinates is shown in Equation (Equation 23).
(23)$$\left\{\begin{array}{c} X=\frac{m_{11}j+m_{12}k+m_{13}}{m_{31}j+m_{32}k+m_{33}}\;\\ Y=\frac{m_{21}j+m_{22}k+m_{23}}{m_{31}j+m_{32}k+m_{33}} \end{array}\right.$$
where, (j,k) denotes the phase plane coordinates, which are obtained directly from the video image; (X,Y,1) denotes the actual spatial Cartesian coordinates; $m_{ab}\left(a,b=1∼3\right)$ denotes the conversion parameters of the two coordinates. Finally, the spatial Cartesian coordinates of any point on the image and the actual distance represented by each pixel are derived from Equation (Equation 23), and then the magnitude of the flow velocity can be calculated.

### 3.4. Calculation of Surface Flow Velocity and Discharge

Assuming that the river surface flow features move along the velocity-measuring line at a distance of *D* in time *T* in the physical coordinate system, which corresponds to the movement of pixels *d* in frames τ in the image coordinate system, then the magnitude of the surface flow velocity vector on the velocity-measuring line is expressed as: (24)$$V=\frac{D}{T}=\frac{d\cdot S_{x}}{\tau \cdot S_{t}}=\tan\theta \cdot \frac{S_{x}}{S_{t}}=\tan\theta \cdot S_{x}\cdot fps$$
where, $S_{x}$ denotes the actual distance represented by each pixel (m/pixel) and fps denotes the camera frame rate (pixel/s).

Thus, the space-time image describes the relationships between the temporal and spatial of the flow features movement in a 2D image, and the measurement of the flow velocity is converted into a measurement of the direction of texture (trajectory) of the space-time image.

After obtaining the river surface flow velocity, the cross-section flow can be calculated according to the flow-area method [29]. The cross-sectional diagram is shown in Figure 7.

The area $A_{i}$ of the section i is: (25)$$A_{i}=\frac{d_{i-1}+d_{i}}{2}b_{i}$$
where, *i* denotes the number of the velocity-measuring vertical line, i=0,1,⋯,n; $d_{i}$ denotes the water depth corresponding to the velocity-measuring vertical line *i*; and $b_{i}$ denotes the width of the section *i*.

The vertical line average velocity is equal to the surface velocity multiplied by the surface velocity coefficient: (26)$$V_{mi}=V_{i}\cdot \alpha$$
where, α denotes the surface velocity coefficient.

The partial average velocity ${\bar{V}}_{i}$ between the two velocity-measuring vertical lines is: (27)$${\bar{V}}_{i}=\frac{V_{m(i-1)}+V_{mi}}{2}$$
the partial average velocity near the shore or stagnant water is: (28)$$\bar{V_{1}}=V_{m1}\cdot \lambda$$
(29)$$\bar{V_{n}}=V_{m(n-1)}\cdot \lambda$$
where, λ denotes the shore velocity coefficient.

The partial discharge is given by: (30)$$q_{i}=\bar{V_{i}}\cdot A_{i}.$$

The total discharge is denoted as: (31)$$Q=\sum_{i=1}^{n}q_{i}.$$

Finally, the average velocity is denoted as: (32)$$\bar{V}=\frac{Q}{A}.$$

## 4. Comparison and Analysis of Results

### 4.1. Experimental Design

According to the Code for liquid flow measurement in open channels GB 50179-2015, hydrologists generally take the measurement results of the propeller current meter as the true value. To better verify the universality, accuracy and real-time, flow comparative tests were conducted between the proposed method and the propeller current meter at the hydrological station in Baoji, Guizhou Province under high flow conditions. Finally, the measurement results of the proposed algorithm are compared and analyzed with the propeller current meter, LSPIV, STIV and FD-DIS-G, mainly for the comparison and analysis of hydrological information such as the vertical line average velocity, average velocity and flow discharge.

The case is a comparative measurement experiment based on the river under high flow conditions at the hydrological station in Baoji, Guizhou. According to the calibration of the hydrological station for many years, the surface velocity coefficient α is 0.88, and the shore coefficient λ is 0.70. Four ground control points, A, B, C, and D, in the same plane is laid on the left bank and right bank, respectively, and EF is the section-line (E is the starting point and F is the end point), the ground control points, velocity-measuring points (the starting distance is 7 m, 12 m, 17 m, 22 m, 27 m, 32 m, 37 m, 42 m) are shown in Figure 8, and the section data are shown in Table 1. Eight velocity-measuring lines of equal length and parallel to the riverbank are laid at the location of the velocity-measuring points in Figure 8, as shown in the red straight line. In this river channel, the river video with a frame rate of 25 fps and a duration of 20 s was captured, and the camera was fixed during the acquisition process without obvious jitter, and the water flow was stable and the effect of wind was negligible. The video frame cutting process yields 499 frames with time interval Δt = 0.04 s and image size 1920×1080 of the river surface image. To ensure the reliability of the measurement results, the vertical line average flow velocity was measured at the same time using LS25-3A propeller current meter, the hydrological data such as partial average velocity, partial discharge and total discharge are shown in Table 2.

Before detecting MOT, it is necessary to perform pre-detection with synthesized images based on the Berlin noise. First, a synthetic space-time image containing Perlin noise with a size of 224×224 pixels is generated in the vertical direction. Then, a total of 900 space-time images with a given angle are generated by rotating the images counterclockwise in $0.1^{\circ }$ steps with the image center as the origin and the angle range of $0^{\circ }$ to $90^{\circ }$.

To verify the detection accuracy of the algorithm, ten images are randomly selected from the synthetic images. The ten randomly selected images are shown in Figure 9 with the given angles of $15.8^{\circ }$, $25.7^{\circ }$, $30.1^{\circ }$, $40.6^{\circ }$, $50.7^{\circ }$, $55.8^{\circ }$, $65.4^{\circ }$, $75.3^{\circ }$, $80.2^{\circ }$, and $84.3^{\circ }$, respectively.

G is the abbreviation for Given, which stands for a given value, and M is the abbreviation for Measured, which stands for a measured value. From Table 3, the values measured by using the new method are in high agreement with the pre-given values, and the maximum absolute error and maximum relative error are $0.7^{\circ }$ and 1.48%, which are less than $1.0^{\circ }$ and 10%, respectively, thus initially indicating the new method is a feasible solution with high measurement accuracy.

### 4.2. Analysis of the Measurement Results

The measurement results of the FMAA method based on the filtering enhancement algorithm are shown in Table 4. Where, the surface flow velocity is calculated by Equation (Equation 23) and the vertical line average flow velocity is obtained by multiplying the surface flow velocity by the surface flow velocity coefficient.

The vertical line average flow velocity, average flow velocity, and flow discharge obtained by the FMAA method based on the filter enhancement algorithm, the LSPIV method, the LGTM method, and the FD-DIS-G method [30] are compared with the propeller current meter results are shown in Table 5, Table 6 and Table 7, and Figure 10.

As shown in Table 5, the analysis of the vertical line average velocity shows that the results obtained by the algorithm proposed are the closest to the true value measured by propeller current meter, and have good agreement compared with the LSPIV, LGTM and FD-DIS-G. From Table 6 and Table 7, it can be obtained that: in the two points near the shore, the absolute and relative errors are 0.37, 0.38, 27.82% and 26.57%, respectively, which are lower than 0.40, 0.47, 30.08% and 32.87% of the FD-DIS-G method; for the fourth measurement point, the error is 29.52%, which is due to the influence of the placement of the velocity-measuring line by the rocks in the water surface; the measurement accuracy of the vertical line average velocity of the remaining measurement points is higher. The absolute and relative errors of the method are only 0.02 and 1.27%, which are lower than the results of FD-DIS-G method (0.04 and 2.55%). The absolute and relative errors are only 1.61 and 1.08%, which are lower than the 4.09 and 2.74% of the FD-DIS-G method. The average velocity and flow discharge are the closest to the true value of the propeller current meter, and the measurement error is the smallest. From Figure 10d, the proposed method takes longer time than the LGTM method and FD-DIS-G method due to the longer time-consuming image enhancement and filtering pre-processing, but the measurement accuracy is improved to a greater extent, so the longer time-consumption is reasonable.

From the experimental results, the proposed method is suitable for the measurement of vertical line average velocity, average velocity and flow discharge of rivers in natural environment, and has good stability and wide applicability, providing support for flood control and scientific management of water resources in river basins.

## 5. Conclusions

In this paper, we have developed a new method for measuring river surface velocimetry using space-time images. The method includes the spatial-domain texture-enhanced algorithm and frequency-domain denoised algorithm. Experimental results show the precision of the proposed method for the measurement of flow velocity and discharge. The proposed method is broadly applicable for river surface features tracking and velocity measurement. In Section 3.2.1, there are four steps for processing the image texture direction, which takes a long time to process from the original image with disorganized texture and blurred direction to the clear texture direction in Figure 3e. Compared with the LSPIV method, the algorithm is much more efficient and accurate; compared with the LGTM method, the running time is about 50 s longer, but the measurement results of velocity and flow discharge are closer to the true value of the propeller current meter, the relative errors of the average velocity and flow discharge calculated by the new method are less than 2% compared with the results of the propeller current meter, and a good balance between accuracy, efficiency and reliability are achieved. The method can meet the actual engineering measurement requirements and is a non-contact continuous river flow monitoring method with lower equipment and labor-cost.

## Figures and Tables

**Figure 1 sensors-23-00955-f001:**
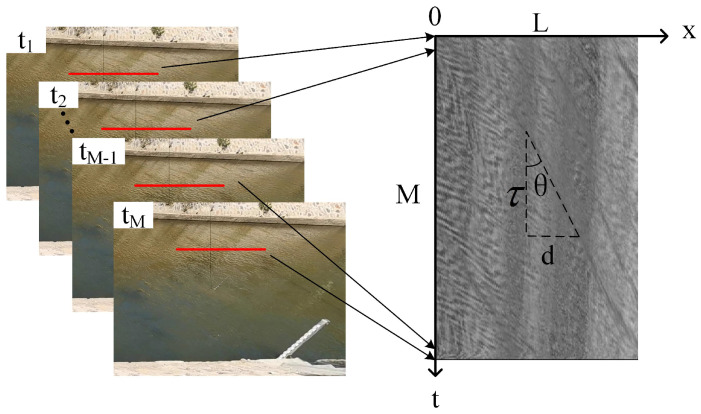
Generation of space-time image.

**Figure 2 sensors-23-00955-f002:**
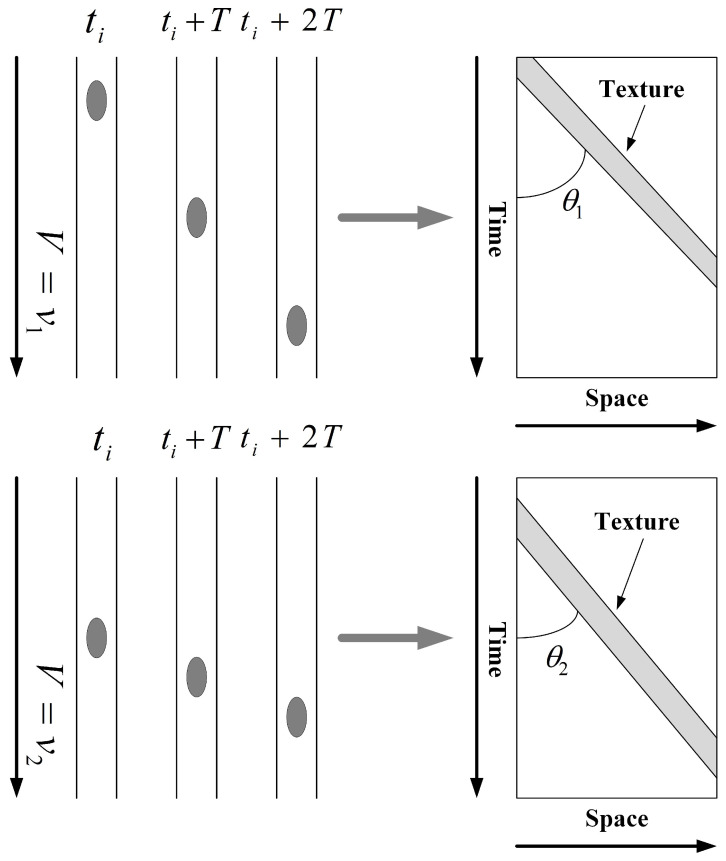
Schematic diagram of the principle of space-time image.

**Figure 3 sensors-23-00955-f003:**
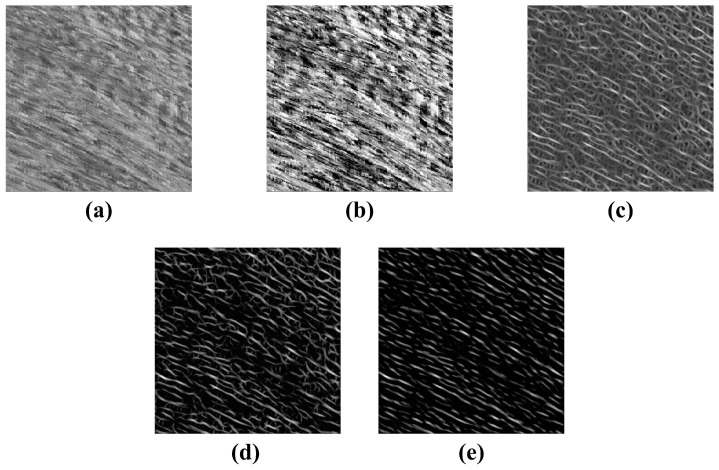
The results of preprocessing of space-time image. (**a**) The original. (**b**) The MSR enhancement of (**a**). (**c**) The result of the Fourth-order Gaussian derivative steerable filter enhancement of (**b**). (**d**) The result of noise filtering function enhancement of (**c**). (**e**) The result of orientation-filtering function enhancement of (**d**).

**Figure 4 sensors-23-00955-f004:**
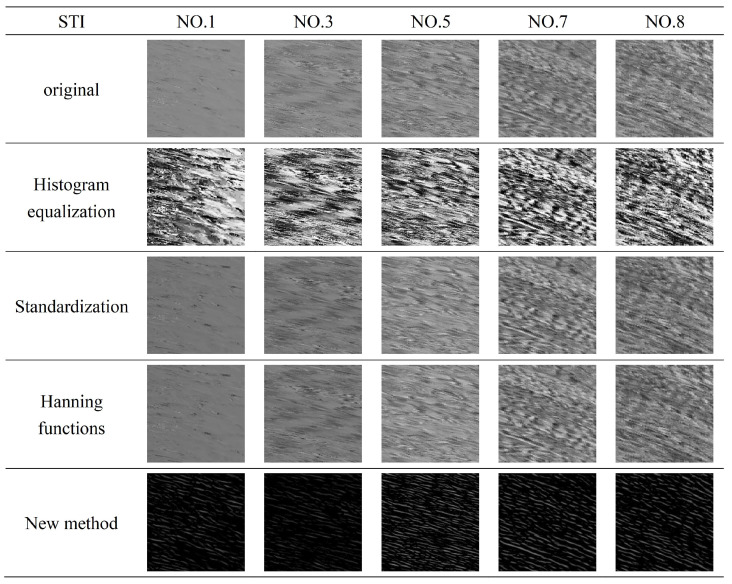
The comparison of preprocessing of space-time image.

**Figure 5 sensors-23-00955-f005:**
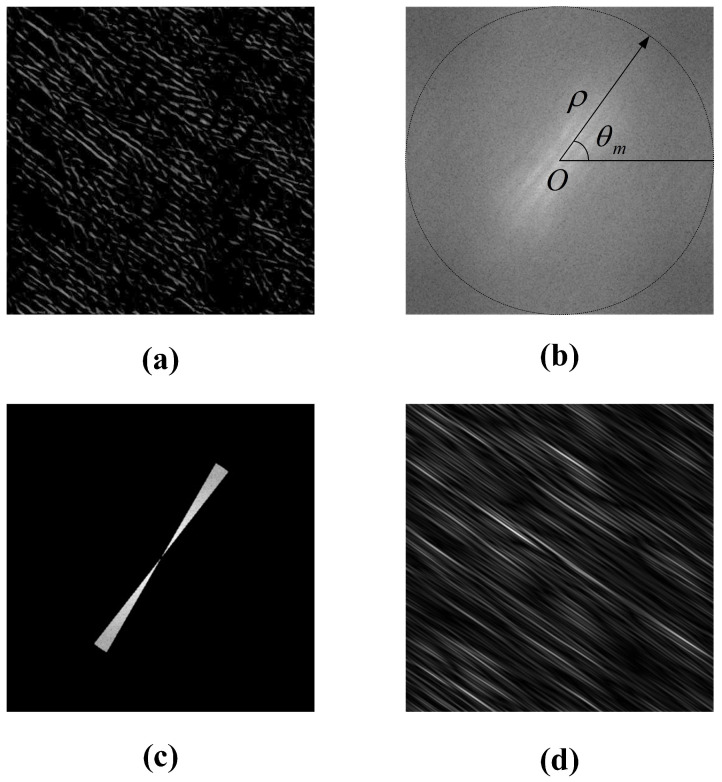
The process of denoising. (**a**) The enhanced STI. (**b**) The polar coordinate system of (**a**). (**c**) The Fan-type masking. (**d**) The denosied STI.

**Figure 6 sensors-23-00955-f006:**
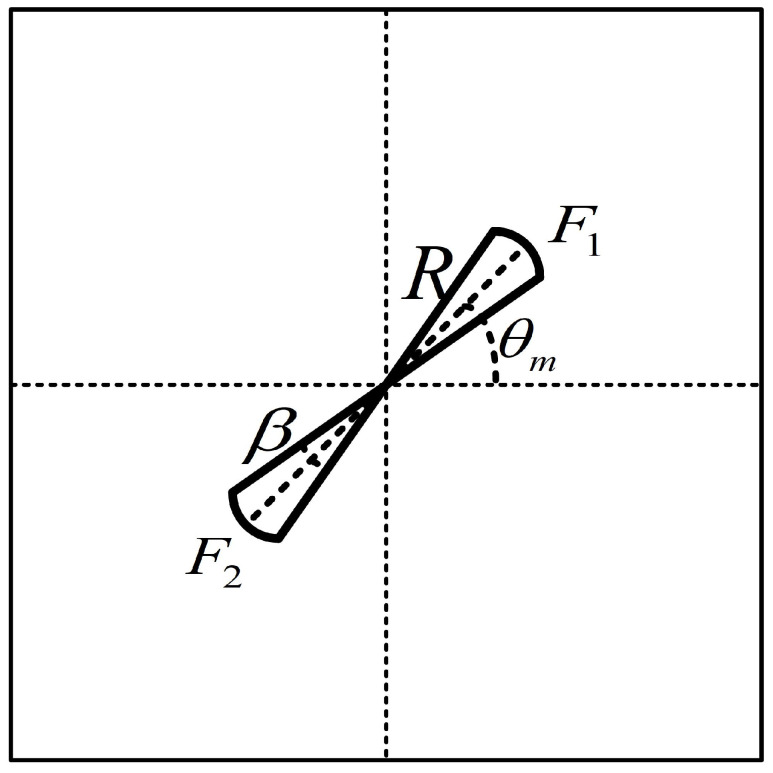
Fan-type filter.

**Figure 7 sensors-23-00955-f007:**
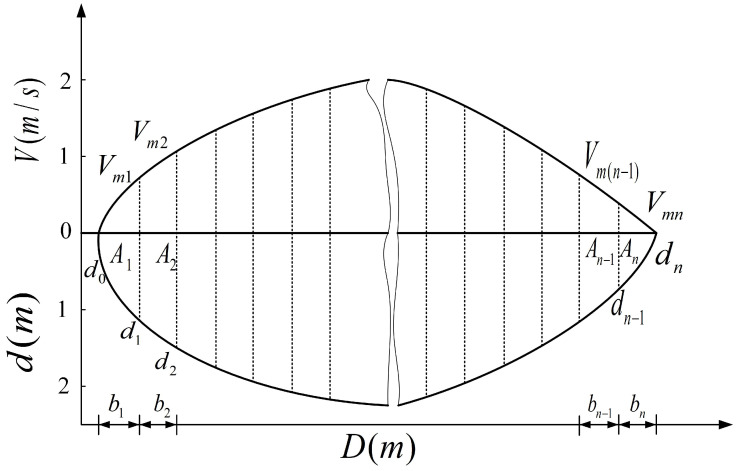
Cross-section.

**Figure 8 sensors-23-00955-f008:**
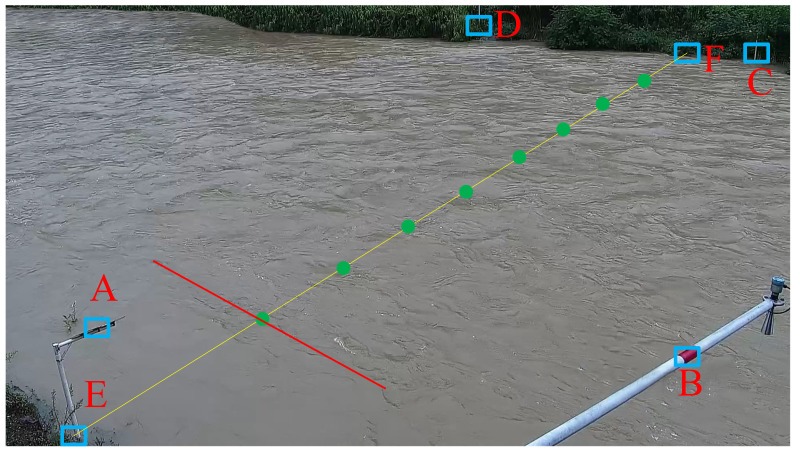
Ground control points (GCPs) and velocity-measuring points.

**Figure 9 sensors-23-00955-f009:**
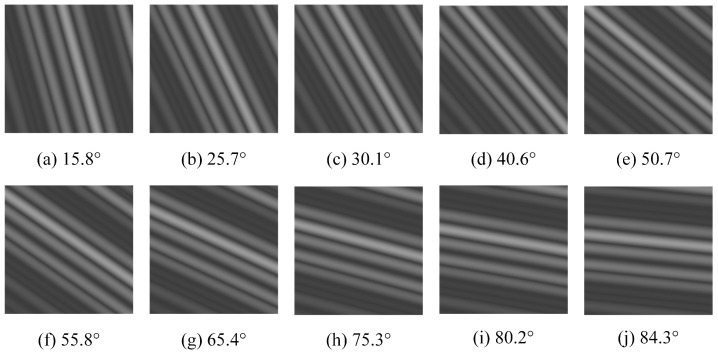
Synthetic images with different angles.

**Figure 10 sensors-23-00955-f010:**
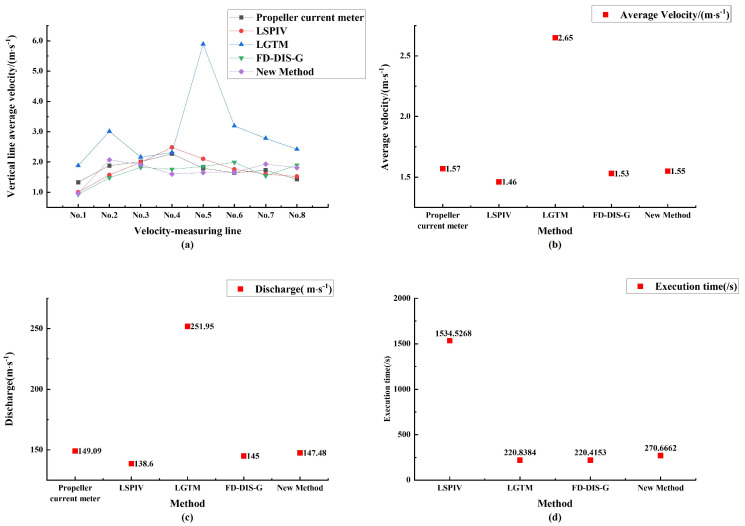
The results of five methods. (**a**) Comparison of vertical line average velocity; (**b**) comparison of average velocity; (**c**) comparison of discharge; and (**d**) comparison of execution time.

**Table 1 sensors-23-00955-t001:** Data of the cross-section.

Distance/(m)	Depth/(m)
1.8	1.37
7	1.90
12	1.75
17	1.72
22	1.84
27	1.88
32	2.12
37	2.33
42	2.47
48.4	1.35
Area/(m^2^)	94.9

**Table 2 sensors-23-00955-t002:** Measurement results of the propeller current meter.

Distance/(m)	Vertical Line Average Velocity/(m · s^−1^)	Partial Average Velocity/(m · s^−1^)	Partial Section Area/(m^2^)	Partial Discharge/ (m^3^ · s^−1^)
0 (left)	0			
0–7		0.93	10.1	9.39
7	1.33			
7–12		1.60	8.97	14.4
12	1.88			
12–17		1.94	8.59	16.7
17	2.01			
17–22		2.14	8.86	19.0
22	2.27			
22–27		2.03	9.18	18.6
27	1.79			
27–32		1.72	10.0	17.2
32	1.64			
32–37		1.68	11.1	18.6
37	1.73			
37–42		1.58	12.3	19.4
42	1.43			
42–51.4		1.00	15.8	15.8
51.4	0			

**Table 3 sensors-23-00955-t003:** Data of the cross-section.

	a	b	c	d	e	f	g	h	i	j
G	$15.8^{\circ }$	$25.7^{\circ }$	$30.1^{\circ }$	$40.6^{\circ }$	$50.7^{\circ }$	$55.8^{\circ }$	$65.4^{\circ }$	$75.3^{\circ }$	$80.2^{\circ }$	$84.3^{\circ }$
M	$16.0^{\circ }$	$26.0^{\circ }$	$30.0^{\circ }$	$40.0^{\circ }$	$50.0^{\circ }$	$56.0^{\circ }$	$65.0^{\circ }$	$75.0^{\circ }$	$80.0^{\circ }$	$84.0^{\circ }$

**Table 4 sensors-23-00955-t004:** Measurement results of the new method.

Velocity-Measuring Line	Surface Velocity/(m · s^−1^)	Vertical Line Average Velocity/(m · s^−1^)	Partial Average Velocity/(m · s^−1^)	Partial Section Area/(m^2^)	Partial Discharge/ (m^3^ · s^−1^)
			0.67	10.10	6.77
No.1	1.09	0.96			
			1.52	8.97	13.63
No.2	2.35	2.07			
			1.99	8.59	17.09
No.3	2.17	1.91			
			1.76	8.86	15.59
No.4	1.82	1.60			
			1.63	9.18	14.96
No.5	1.88	1.65			
			1.65	10.00	16.50
No.6	1.88	1.65			
			1.79	11.10	19.87
No.7	2.19	1.93			
			1.87	12.30	23.00
No.8	2.06	1.81			
			1.27	15.80	20.07
Total discharge/(m^3^ · s^−1^): 147.48					
Total cross-section area/(m^2^): 94.90					
Average velocity/(m · s^−1^): 1.55					

**Table 5 sensors-23-00955-t005:** Comparison of measurement results of five methods.

Method	Vertical Line Average Velocity/(m · s^−1^)								Average Velocity/(m · s^−1^)	Discharge/(m^3^ · s^−1^)
	No.1	No.2	No.3	No.4	No.5	No.6	No.7	No.8		
Current meter	1.33	1.88	2.01	2.27	1.79	1.64	1.73	1.43	1.57	149.09
LSPIV	1.00	1.57	1.99	2.48	2.10	1.76	1.60	1.52	1.46	138.60
LGTM	1.88	3.01	2.16	2.31	5.89	3.19	2.78	2.42	2.65	251.95
FD-DIS-G	0.93	1.48	1.82	1.76	1.85	1.99	1.54	1.90	1.53	145.00
New Method	0.96	2.07	1.91	1.60	1.65	1.65	1.93	1.81	1.55	147.48

**Table 6 sensors-23-00955-t006:** Absolute errors of the vertical line average velocity of four methods.

Method	Absolute Errors/(m · s^−1^)								Average Velocity/(m · s^−1^)	Discharge/(m^3^ · s^−1^)
	No.1	No.2	No.3	No.4	No.5	No.6	No.7	No.8		
LSPIV	0.33	0.31	0.02	0.21	0.31	0.12	0.13	0.09	0.11	10.49
LGTM	0.55	1.13	0.15	0.04	4.10	1.55	1.05	0.99	1.08	102.86
FD-DIS-G	0.40	0.40	0.19	0.51	0.06	0.35	0.19	0.47	0.04	4.09
New Method	0.37	0.19	0.10	0.67	0.14	0.01	0.20	0.38	0.02	1.61

**Table 7 sensors-23-00955-t007:** Relative errors of the vertical line average velocity of four methods.

Method	Relative Errors/(%)								Average Velocity/(%)	Discharge/(%)
	No.1	No.2	No.3	No.4	No.5	No.6	No.7	No.8		
LSPIV	24.81	16.49	1.00	9.25	17.32	7.32	7.51	6.29	7.01	7.04
LGTM	41.35	60.11	7.46	1.76	229.05	94.51	60.69	69.23	68.79	68.99
FD-DIS-G	30.08	21.28	9.45	22.47	3.35	21.34	10.98	32.87	2.55	2.74
New Method	27.82	10.11	4.98	29.52	7.82	0.61	11.56	26.57	1.27	1.08

## Data Availability

Not applicable.

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
