# Peer review of "Velocity Vector Estimation of Two-Dimensional Flow Field Based on STIV"

_sensors, 2023, doi:10.3390/s23020955_

Round 1

Reviewer 1 Report

Based on a comprehensive investigation of the relevant methods, the paper proposed an integrated method for river discharge monitoring.

There are some points:

1) The paper shows the advantages and disadvantages of verious methods in the "Introduction" section. However, what are the advantages of the STIV method over other methods? Why is this method used in the paper? It is suggested to explain in section 2.

2) In section 3.2.1, the titles of steps (iii) and (iiii) are identical. Why to use the method of step (iii) firstly, and then use that of the step (iiii)? How is the filtering steps considered?

3) In Figure 3, not all the textures of the images are clear. How does Figure 3(a), for example, compare with other methods?

4) The frequency domain filtering is highly sensitive to the parameters of fan-type filter. How are the parameters considered in this filter?

5) The measurement comparison results show that the new method has high accuracy, but the time-consumption is relatively long. Have the study calculated which step in the whole processing causes the overall time to be long, and whether some improvements can be performed?

Author Response

Dear reviewer,

Thank you very much for giving us an opportunity to revise our manuscript and for the comments on our manuscript entitled "Velocity Vector Estimation of Two-dimensional Flow Field based on STIV" (Manuscript ID: sensors-2099269). Those comments are constructive for revising and improving our paper. We have studied the comments carefully and made corrections which we hope meet with approval and publication in this journal. The primary corrections are in the manuscript, and the responses to the reviewers' comments are as follows:

(All revised sections are highlighted in red, and new additions are highlighted in blue. We would like to submit it for your consideration.)

Comments: 1) The paper shows the advantages and disadvantages of various methods in the "Introduction" section. However, what are the advantages of the STIV method over other methods? Why is this method used in the paper? It is suggested to explain in section 2.

Response: In section 2. Outline of the Space-Time Image Velocimetry, we made minor additions “Among the image-based methods, the Space-Time Image Velocimetry (STIV) is considered a powerful tool for obtaining hydrologic information. The STIV is a time-averaged velocity measurement method, which takes river surface images as the analysis object and takes the single pixel-wide velocity-measuring line as the analysis region, and detects the MOT in a generated space-time image to obtain one-dimensional velocities of the water surface, which has higher spatial resolution and computational efficiency and is more suitable for shore-based systems with a small depression angle. Among five methods of STIV, the FMAA method converts the complex MOT detection in the spatial domain into detecting the MOS in the frequency domain, which simplifies the complex convolution or gradient operation in the spatial domain and provides higher robustness to random noise and interfering textures in space-time images, and the computational efficiency is improved accordingly.”

Comments: 2) In section 3.2.1, the titles of steps (iii) and (iiii) are identical. Why to use the method of step (iii) firstly, and then use that of the step (iiii)? How is the filtering steps considered?

Response: The titles of steps (iiii) have been revised. After the fourth-order Gaussian derivative steerable filtering enhancement process, the low-contrast texture is enhanced, but the noise is also enhanced simultaneously, so we first use step (iii) of the noise filtering function to filter out the noise, giving higher weights to the texture-related pixels and lower weights to the noise-related pixels, to highlight the texture direction information. After the above three steps of enhancing and suppressing noise, the next step is to use the orientation filtering function to further reduce any other noise that may remain.

Comments: 3) In Figure 3, not all the textures of the images are clear. How does Figure 3(a), for example, compare with other methods?

Response: In Figure 3, (a) the Original. (b) the MSR enhancement of (a). (c) the result of the Fourth-order Gaussian derivative steerable filter enhancement of (b). (d) the result of Noise filtering function enhancement of (c). (e) the result of Orientation-filtering function enhancement of (d). Figure 3(a) after four steps of enhancement and filtering, the texture orientation information is clearer and with lower noise, as shown in Figure 3(e).

Comments: 4) The frequency domain filtering is highly sensitive to the parameters of fan-type filter. How are the parameters considered in this filter?

Response: A sensitivity analysis of the parameter settings of the fan-type filter in the frequency domain was done in Reference[28], and it was found that the filter can effectively filter out the noise when the radius was one-fourth of the size of the image and the angle was 5.3 degrees. Since section 3.2.1 Space-time image filtering and enhancement has reduced the noise to a certain extent, the angle is set to 4 degrees in this paper.

Comments: 5) The measurement comparison results show that the new method has high accuracy, but the time-consumption is relatively long. Have the study calculated which step in the whole processing causes the overall time to be long, and whether some improvements can be performed?

Response: In section 3.2.1 Space-time image filtering and enhancement, there are four steps for processing the image texture direction, which takes a long time to process from the original image with disorganized texture and blurred direction to the clear texture direction in Figure 3(e). Compared with the LSPIV method, the algorithm is much more efficient and accurate; compared with the LGTM method, the running time is about 50 seconds longer, but the measurement results of velocity and flow discharge are closer to the true value of the propeller current meter, the relative errors of the average velocity and flow discharge calculated by the new method are less than 2% compared with the results of the propeller current meter, and a good balance between accuracy, efficiency and reliability are achieved.

Reviewer 2 Report

Comments on Review:

1. In the section of 2. Outline of the Space-Time Image Velocimetry, the introduction of the Space-Time Image Velocimetry is not clear and does not highlight the key content. It is suggested that the author improve the explanation.

2. In the section of 3.1. Generation of space-time images, the author expresses as the MOT θ1>θ2, but we can see it in Figure 2 θ1<θ2, contrary to the previous statement, please check.

3. In the section of 3. Principle of the Space-Time Image Velocimetry, many paragraphs are indented incorrectly.

4. In equation 4, wk is not defined, please state if no definition is required.

5. In equation 5, please check whether there are too many symbols, if not please explain.

6. In the section of 3.2.1. Space-time image filtering and enhancement, (iii) Noise filtering function is the same as (iiii) Noise filtering function, please check for changes.

7. In the section of 4.1. Experimental design, the author describes the establishment of four control points ABC and D, but it does not appear in the later content. Please explain the purpose of the establishment of control points ABC and D.

8. In the section of 4.2. Analysis of the measurement results, the vertical line is represented by number 1-8, which does not appear in Section 4.1. Moreover, in the later content, the reference to the measurement points are not clear, so it is suggested that the author unify the description of the measurement line.

9. In order to highlight the advantages of the new method, the author compares many hydraulic parameters, but the meaning of the parameters is not explained. It is suggested that the author give a theoretical explanation.

Author Response

Dear reviewer,

Thank you very much for giving us an opportunity to revise our manuscript and for the comments on our manuscript entitled "Velocity Vector Estimation of Two-dimensional Flow Field based on STIV" (Manuscript ID: sensors-2099269). Those comments are constructive for revising and improving our paper. We have studied the comments carefully and made corrections which we hope meet with approval and publication in this journal. The primary corrections are in the manuscript, and the responses to the reviewers' comments are as follows:

(All revised sections are highlighted in red, and new additions are highlighted in blue. We would like to submit it for your consideration.)

Comments: 1) In the section of 2. Outline of the Space-Time Image Velocimetry, the introduction of the Space-Time Image Velocimetry is not clear and does not highlight the key content. It is suggested that the author improve the explanation.

Response: In section 2. Outline of the Space-Time Image Velocimetry, we made minor additions “Among the image-based methods, the Space-Time Image Velocimetry (STIV) is considered a powerful tool for obtaining hydrologic information. The STIV is a time-averaged velocity measurement method, which takes river surface images as the analysis object and takes the single pixel-wide velocity-measuring line as the analysis region, and detects the MOT in a generated space-time image to obtain one-dimensional velocities of the water surface, which has higher spatial resolution and computational efficiency and is more suitable for shore-based systems with a small depression angle. Among five methods of STIV, the FMAA method converts the complex MOT detection in the spatial domain into detecting the MOS in the frequency domain, which simplifies the complex convolution or gradient operation in the spatial domain and provides higher robustness to random noise and interfering textures in space-time images, and the computational efficiency is improved accordingly.”

Comments: 2) In the section of 3.1. Generation of space-time images, the author expresses as the MOT θ12, but we can see it in Figure 2 θ12, contrary to the previous statement, please check.

Response:  in Figure 2, the MOT θ12 has been revised.

Comments: 3) In the section of 3. Principle of the Space-Time Image Velocimetry, many paragraphs are indented incorrectly.

Response: The indentation of paragraphs have been revised.

Comments: 4) In equation 4, wk is not defined, please state if no definition is required.

Response: In equation 4, wk has been defined.

Comments: 5) In equation 5, please check whether there are too many symbols, if not please explain.

Response: In equation 5, there are not too many symbols.  $\theta^*(x, y)$ indicates the orientation of the pixel when it has the maximum response. $h_\theta(x, y)$ denotes the rotated steerable filter bank [25].

Comments: 6) In the section of 3.2.1. Space-time image filtering and enhancement, (iii) Noise filtering function is the same as (iiii) Noise filtering function, please check for changes.

Response: The titles of steps (iiii) have been revised with the orientation filtering function.

Comments: 7) In the section of 4.1. Experimental design, the author describes the establishment of four control points A、B、C and D, but it does not appear in the later content. Please explain the purpose of the establishment of control points A、B、C and D.

Response: The four ground control points A, B, C, and D (GCPs) are set to perform ground calibration to convert the motion vectors in the flow field from the image coordinate system to the world coordinate system to find the actual distance represented by each pixel from equation (23), and then the magnitude of the flow velocity can be calculated, as shown in Section 3.3 Calibration.

Comments: 8) In the section of 4.2. Analysis of the measurement results, the vertical line is represented by number 1-8, which does not appear in Section 4.1. Moreover, in the later content, the reference to the measurement points are not clear, so it is suggested that the author unify the description of the measurement line.

Response: In the section of 4.1 Experimental design, we made minor additions “eight velocity-measuring lines of equal length and parallel to the riverbank are laid at the location of the velocity-measuring points in Figure 8, as shown in the red straight line.”

Comments: 9) In order to highlight the advantages of the new method, the author compares many hydraulic parameters, but the meaning of the parameters is not explained. It is suggested that the author give a theoretical explanation.

Response: The hydrological parameters are explained in detail in section 3.4 Calculation of surface flow velocity and discharge, and these hydrological parameters fully comply with the standard in reference [29]. The vertical line average velocity is equal to the surface velocity multiplied by the surface velocity coefficient. The partial average velocity is the average velocity of the partial cross-sectional area between the two velocity-measuring vertical lines, as well as the average velocity of the parts between the shore or stagnant water and the velocity-measuring vertical lines at both ends of the section.

Round 2

Reviewer 1 Report

Thank you for your effort in improving the manuscript. In any case, I have two more suggestions:

1) The responses can be properly added to the manuscript to better facilitate the reader's understanding.

2) The English language needs further improvement.

Author Response

Dear Reviewer,

Thank you very much for giving us an opportunity to revise our manuscript and for the comments on our manuscript entitled "Velocity Vector Estimation of Two-dimensional Flow Field based on STIV" (Manuscript ID: sensors-2099269). I have finished my revisions and uploaded the revised version together with my responses to the reviewers. Those comments are constructive for revising and improving our paper. We have studied the comments carefully and made corrections which we hope meet with approval and publication in this journal. 

(All revised sections are highlighted in red, and new additions are highlighted in blue. We would like to submit it for your consideration.)

Comments: 1) The responses can be properly added to the manuscript to better facilitate the reader's understanding.

Response: I have added the responses to the manuscript to facilitate the reader's understanding.

Comments: 2) The English language needs further improvement.

Response:  We tried our best to improve the manuscript and made some changes to the manuscript. These changes will not influence the content and framework of the paper. All revised sections are highlighted in red, and new additions are highlighted in blue.
